# IMPROVING ML ATTACKS ON LWE WITH DATA REPETITION AND STEPWISE REGRESSION

## ABSTRACT

ML attacks on Learning with Errors (LWE) with binary or small secrets only succeed on LWE settings with very simple secrets. For example, they can recover secrets with up to three non-zero bits when models are trained on not-reduced LWE data, and three non-zero bits in the "cruel region (Nolte et al., 2024)" when BKZ pre-processing is applied. We show that larger training sets and the use of repeated examples in the training data allow the recovery of denser secrets. We empirically observe a power-law relationship between model based attempts to recover the secrets, dataset size and repeated examples. We introduce a stepwise regression technique to recover the "cool bits" of the secret. Overall, these techniques allow for the recovery of denser binary secrets: up to Hamming weight 70 (and 8 cruel bits) for dimension 256 $\log_2 q = 20$ and 75 (and 7 cruel bits) for dimension 512 $\log_2 q = 41$ (vs 33 and 63 Hamming weight and 3 cruel bits in previous works). We also demonstrate our methods' effectiveness on denser ternary secrets, showing a substantial improvement over prior work.

## 1 INTRODUCTION

Most current public-key cryptosystems, used to secure many online interactions, are susceptible to attacks based on Shor's algorithm (Shor, 1994), a fast method for integer factorization on a quantum computer. To counter this, a new class of systems, known as post-quantum cryptography (PQC), have been developed and are currently being standardized (Chen et al., 2022). Assessing their potential weaknesses and limitations is an active field of research.

Many PQC systems are based on a hard math problem known as Learning With Errors (LWE) (Regev, 2005). The security of LWE stems from the hardness of recovering a secret vector of integers modulo $q$ from its noisy dot products with random vectors: find $\mathbf{s} \in \{0, \ldots, q-1\}^n$, given $p$ pairs $(\mathbf{a}_i, b_i = \mathbf{a}_i \cdot \mathbf{s} + \epsilon_i \mod q)_{i \in \{1\ldots p\}}$, with $\mathbf{a}_i$ randomly sampled from a uniform distribution over $\{0, \ldots, q-1\}^n$, and $\epsilon_i$ sampled from a discrete Gaussian distribution with low standard deviation (typically, 3 or 3.2).

When the coordinates of $\mathbf{s}$ are sampled over the whole hypercube, LWE is provably hard for large values of $n$, and not too large moduli $q$. Versions of LWE using *sparse* and/or binary, ternary or small secrets (i.e. most of the coordinates of $\mathbf{s}$ are zero, non-zero coordinates are 1, 1 or $-1$, or small integers, respectively) have been proposed. These alternatives are attractive, because they accelerate encryption and decryption, and allow for *homomorphic encryption* – performing computations on encrypted data, without decrypting them. Reductions exist that suggest these simpler alternatives may offer the same security guarantee as the full-fledged version of LWE, but the potential weakness of sparse and small, for practical values of $n$ and $q$, is under debate. This is the subject of this paper.

SALSA, a transformer-based ML attack on LWE with small and sparse secrets was proposed in prior works (Wenger et al., 2022; Li et al., 2023b;a; Stevens et al.). SALSA uses LWE pairs sharing the same secret, $(\mathbf{a}, b = \mathbf{a} \cdot \mathbf{s} + \epsilon \mod q)$, to train a transformer to predict $b$ from $\mathbf{a}$. Once the model achieves better than random predictive performance, secret coordinates can be recovered by comparing model predictions for close values of $\mathbf{a}$. Later works (Li et al., 2023b;a; Stevens et al.) introduced a pre-processing step: LWE pairs, created from a small eavesdropped sample, are first reduced by lattice algorithms (BKZ (Schnorr, 1987; Chen & Nguyen, 2011)) and then used to train the transformer. BKZ reduction causes the last coordinates of $\mathbf{a}$ (the cool region) to have low variance,

while the $c$ first coordinates (the cruel region) remain unreduced (Nolte et al., 2024). These works show that BKZ reduction enables the recovery of secrets with larger Hamming weight $h$ (number of non-zero secret bits) (Li et al., 2023b;a; Stevens et al.).

However, ML attacks on LWE are constrained by the model's ability to learn the modular dot product. In particular, models tend to struggle when the dot product $\mathbf{a} \cdot \mathbf{s}$ grows large (Li et al., 2023a). This happens when there are more than three non-zero bits in the unreduced (cruel) region of the secret (Wenger et al., 2025). This limits the success of ML attacks as the Hamming weights of secrets increases. For example, for $n = 512$, with $\log_2 q = 28$, after BKZ reduction, the cruel region is the first 224 bits. Therefore, most secrets with $h \geq 10$ will have 4 cruel bits or more, and will not be recoverable.

In this paper, we explore methods that allow ML attacks to recover secrets with more than 3 cruel bits. We explore the impact of larger training sets and repeated examples. We also introduced two new methods for recovering cool bits, based on stepwise regression (Efroymson, 1960). Overall, **our contributions** are:

- training on larger datasets and repeated examples, enabling the recovery of binary and ternary secrets with up to 8 cruel bits,
- replacing the linear regression used to recover cool bits (Wenger et al., 2025) by stepwise regression,
- exploring secret recovery as a function of training set size, repetition, secret Hamming weight and number of cruel bits, in four LWE settings, through 400 million synthetic data samples,
- defining an empirical scaling power law relating secret recovery to data amount and data repetitions.

Overall, our attack recovers secrets with larger Hamming weights than previous works (Table 1), and demonstrates that the limitation of ML attacks to secrets with 3 cruel bits can be overcome.

Table 1: **Largest Hamming weights recovered (two settings).**

| Settings | Ours | SALSA | C&C | Settings | Ours | SALSA | C&C |
|---|---|---|---|---|---|---|---|
| n=256 $\log_2 q = 12$ | **14** | 8 | 12 | n=256 $\log_2 q = 12$ | **12** | 9 | - |
| n=256 $\log_2 q = 20$ | **70** | 33 | - | n=256 $\log_2 q = 20$ | **55** | 24 | - |
| n=512 $\log_2 q = 28$ | **12** | - | 12 | n=512 $\log_2 q = 28$ | **10** | - | - |
| n=512 $\log_2 q = 41$ | **75** | 63 | 60 | n=512 $\log_2 q = 41$ | **75** | 66 | - |
| Binary secret | | | | Ternary secret | | | |

## 2 RELATED WORK

SALSA (Wenger et al., 2022) is the first ML attack on LWE. It uses a shallow sequence-to-sequence transformer (Vaswani et al., 2017), with shared layers (Dehghani et al., 2019), trained on 2 million unreduced LWE pairs, and recovers secrets from the trained model with a basic distinguisher. Limited to dimensions up to 128 and Hamming weight 3, it is proof of concept: all secrets recovered by SALSA could be found with exhaustive search. It also requires millions of eavesdropped LWE pairs with the same secret, an unrealistic assumption.

PICANTE (Li et al., 2023b) introduces pre-processing. It only requires $4n$ eavesdropped LWE pairs (a realistic assumption) which are sampled to form $n \times n$ matrices $\mathbf{A}$, and reduced by BKZ to produce matrices $\mathbf{RA}$ with a low standard deviation of their entries. Applying the same transformation to $\mathbf{b} = \mathbf{A} \cdot \mathbf{s} + \epsilon$ yields reduced LWE pairs with the same secret, but increased noise. A transformer is then trained from 4 million reduced pairs, that can recover sparse binary secrets with $h = 31$ for $n = 256$ and $\log_2 q = 23$, and $h = 60$ for $n = 350$ and $\log_2 q = 32$.

VERDE (Li et al., 2023a) improves reduction techniques and distinguishers, recovering secrets with $h = 63$ for $n = 512$ and $\log_2 q = 41$. It also extends the recovery mechanism to ternary and small secrets, demonstrating that ternary secrets are not more secure than binary. Finally, it provides a theoretical explanation of the role of reduction: models struggle to learn modular additions that exceed a multiple of the modulus. FRESCA (Stevens et al.) introduces angular embeddings, an architecture for modular arithmetic, which halves the length of inputs, allowing secret recovery for $n = 1024$ and $\log_2 q = 50$.

The difficulty of learning modular addition, first observed by (Palamas, 2017), was further studied by (Saxena et al., 2024). They show that noiseless modular addition of many integers can be learned by adding a regularizer to the loss introduced in FRESCA and adding sparser examples to the training data, in a manner of curriculum learning.

(Nolte et al., 2024) offer a different perspective on BKZ reduction. They observe that the reduction of $\mathbf{a}$ is concentrated to the last coordinates (the cool region), while the $c$ first (the cruel region) stay unreduced. Thanks to the reduction, the cool bits of the secret are easy to recover once the cruel bits known. Therefore, recovering secrets from reduced LWE pairs amounts to discovering their cruel bits, and ML attacks only recover secrets with up to three cruel bits. The paper also proposes a non-ML attack, where all possible cruel bits are enumerated, and the cool bits then guessed. It performs well for small dimensions and reduced data (Wenger et al., 2025), but scales exponentially with the Hamming weight to recover, and requires large samples of LWE pairs for cool bit recovery. (Wenger et al., 2025) provides a comparison between SALSA, Cool and Cruel and classic attacks on LWE for $n = 256$ and $1024$.

Training from repeated examples was proposed by (Charton & Kempe, 2024). They show that multiplication modulo 67, a task that transformer cannot usually learn when trained from large datasets of single-use examples, can be learned with $100\%$ accuracy when models are trained from smaller sets of repeated data. (Saxena et al., 2024) demonstrate a similar effect for modular addition. These papers are the original inspiration for the methods presented in section 4.

## 3 EXPERIMENTAL SETTINGS

**LWE settings.** We consider four sets of parameters for LWE problems (see Table 14 in Appendix A): $n = 256$, $\log_2 q = 12$, $n = 512$, $\log_2 q = 28$, $n = 256$, $\log_2 q = 20$, and $n = 512$, $\log_2 q = 41$. In the two first settings, the low value of the modulus complicates BKZ reduction, and the size of the cruel region is large. The two last settings allow for more reduction, less cruel region, higher recoverable Hamming weights, but a lot of cool bits. In all four settings, the standard deviation of error is 3. These settings are also used in prior work (Li et al., 2023a; Nolte et al., 2024).

We choose $n$ for which running lattice BKZ reduction is possible with reasonable resources but running brute force attacks is not feasible. For $n = 256$, recovering a $h = 14$ secret takes $10^{22}$ attempts, each requiring thousands of operations. This would take several months on the fastest current supercomputer ($10^{18}$ FLOPS), at the frontier of current brute force capability. For $n = 512$, recovering a 75-bit secret requires $10^{91}$ attempts, which is far beyond any brute-force attack capability.

**Data generation and reduction.** We implement the AI-based attack described in Fresca (Stevens et al.; Wenger et al., 2025). Starting from $4n$ random uniform vectors $\mathbf{a}_i \in \mathbb{Z}_q^n$, we sample $m \leq n$ without replacement ($m = 0.875n$ usually), and stack them in a matrix $\mathbf{\Lambda} = \begin{bmatrix} 0 & q \cdot \mathbf{I}_n \\ \omega \cdot \mathbf{I}_m & \mathbf{A} \end{bmatrix}$.

The matrix $\mathbf{\Lambda}$ is then reduced, using the interleaved BKZ2.0 and flatter (Nolte et al., 2024), yielding a $(n + m) \times n$ matrix $\mathbf{R\Lambda}$ which has entries with a smaller variance. For a given secret $\mathbf{s}$, the LWE pairs $(\mathbf{A}, \mathbf{b} = \mathbf{A} \cdot \mathbf{s} + \epsilon)$ are then transformed by $\mathbf{R}$ to provide $n + m$ reduced LWE pairs $(\mathbf{RA}, \mathbf{Rb})$, with the same secret, but a larger error $\epsilon$ (the parameter $\omega$, set to 10 (Li et al., 2023b) controls the trade-off between reduction and error amplification). This process is repeated to generate the train and test sets. We utilized CPUs with 750 GB of RAM and around 42M CPU hours to generate the BKZ-reduced datasets, please refer to Appendix A for a more detailed breakdown.

After reduction, the variance of the last $n - c$ columns of $\mathbf{RA}$ is reduced to less than half the standard deviation of an uniform distribution, $\frac{q}{\sqrt{12}}$. The first $c$ columns, the cruel region, stay unreduced (Nolte et al., 2024). For a given $n$, the size of cruel region $c$ decreases as the modulus $q$ increases. Table 14 provides the reduction parameters for the four settings used in this paper.

**Synthetic data.** BKZ reduction uses a large amount of resources: for $n = 512$, it takes about one minute on one CPU to produce one reduced-LWE pair. For our experiments with large training sets (Sections 4), we created 400 million reduced pairs for $n = 256$ and $\log_2 q = 20$. For the other settings, we only created between 4 and 60 million reduced pairs using BKZ reduction. We also generated, for all setting, 400 million synthetic reduced vectors $\mathbf{a}$, with the same coordinate variance as the reduced pairs (i.e. $c$ unreduced coordinates, and $n - c$ reduced with the same amount of

reduction as BKZ reduction). The results from section 6 indicate that secret recovery rates are the same for models trained on BKZ-reduced and synthetic data. An additional discussion can be found in Appendix B. We utilized around 1,000 CPU hours to generate the synthetic datasets.

**Model architecture.** The reduced data is used to train an encoder-only transformer with 4 layers, an embedding dimension $d = 256$ (for settings with the larger values of $q$) and 512 (for the harder settings with lower $q$) and a ratio of dimension to heads of $d/h = 64$. Each coordinate $a_i$ of the input vector $\mathbf{a}$ is encoded by the angular embedding introduced in (Stevens et al.), as the point $(\cos(\frac{2\pi a_i}{q}), \sin(\frac{2\pi a_i}{q})) \in \mathbb{R}^2$ and mapped onto $d$-dimensional space by a learnable linear layer. Two learnable positional embeddings are added to the input: an absolute position embedding (from 1 to $n$), and a binary embedding indicating whether the current position is in the cruel or cool region. Overall, the input vector for coordinate $a_i$ of $\mathbf{a}$ is $\text{Emb}(a) = (\text{Emb}_1(a_i) + \text{Emb}_2(i) + \text{Emb}_3(1_{i \leq c}))$, with $\text{Emb}_1$, $\text{Emb}_2$, $\text{Emb}_3$ $2 \times d$, $n \times d$ and $2 \times d$ matrices. $\text{Emb}_3$, the cool and cruel embedding, is an improvement we introduce which allows for better cruel bit recovery rates over previous works. For $n = 256 \log_2 q = 20$, with 400 million samples, it boosts the recovery rate of secrets with 4 cruel bits from $2/5$ (without the embedding) to $5/5$ (see Appendix C for additional details).

As in FRESCA (Stevens et al.), the transformer output, a sequence of $n$ vectors in $\mathbb{R}^d$, is max-pooled, and processed by a linear layer of size $d \times 2$. Its output $(x, y)$ is decoded as the integer $p$ such that $(\cos(\frac{2\pi p}{q}), \sin(\frac{2\pi p}{q}))$ is closest to $(x, y)$.

**Model training.** All experiments run on 1 V100 GPUs with 32 GB of memory, for 2 billion examples at most. Training time is around 5 days. The model is trained to minimize the mean-square error (MSE) between model predictions and correct answers, using the Adam optimizer (Kingma & Ba, 2015) with mini-batches of 256. In theory, all model predictions should lie on the unit circle, but prior work (Saxena et al., 2024) observed that they tend to drift towards the origin at the beginning of training. They proposed to add the penalty $\alpha \left(r^2 + \frac{1}{r^2}\right)$, with $r^2 = x^2 + y^2$ and $\alpha = 0.1$, to the MSE loss. At the beginning of training, when the model makes random predictions of $b$, the MSE loss has a local minimum at the origin. Unfortunately, the loss at $(0, 0)$ is independent of $b$, therefore the closer predictions are to the origin, the less the model learns. To prevent such a collapse, we consider a more general penalty of the form $P(r) = \alpha r^2 + \beta/r^2$ to the MSE loss. For the remaining sections of the paper, we set $\alpha = \beta = 0.1$, and we defer the discussion on the best parameters with an ablation study across different settings to Appendix D.

**Secret recovery.** At the end of each epoch (conventionally defined as 2.5 million training examples), an attempt is made to recover the secret. The distinguisher introduced in Li et al. (2023a) is run on 1000 reduced LWE examples, and ranks the $c$ cruel columns of the secret by their likeliness of being different from zero. The rank is then used to generate $15,000$ model based attempts of the cruel bits. For each of them, the cool bits are then estimated using stepwise regression 5, producing a secret guess that is evaluated using the statistical test defined at the end of section 4.3 of PICANTE Li et al. (2023b). If the correct secret is discovered, the process stops, else another training epoch is run.

## 4 RECOVERING HIGHER HAMMING WEIGHTS WITH LARGE SETS OF REPEATED EXAMPLES

Prior work suggested that ML attacks on LWE samples cannot recover secrets with $h > 3$ when trained on non-reduced data, or more then 3 cruel bits when trained on reduced data (Nolte et al., 2024; Wenger et al., 2025). In this section, we demonstrate that larger training sets and repeated examples allow to recover more than 3 unreduced secret bits.

We first consider models trained on **non-reduced data**, as in the original SALSA paper. This setting is the cleanest possible, as there can be no side effects due to BKZ reduction, cool bits, or noise amplification. We experiment with $n = 64$ and $\log_2 q = 20$, and secrets with $3 \leq h \leq 6$, in the presence and absence of noise. For each value of $h$, we train models on 3 different secrets, and report success if one is recovered at least.

As in SALSA, secrets with Hamming weight 3 are always recovered, even when the model is trained without repetition on one million sample only. Table 2 presents our findings for $h = 4$ to 6. Secrets with $h = 4$ are recovered without repetition for large data budgets, but repetition enables recovery even for small data budgets. For larger Hamming weights require large data budgets *and* repetition.

These results, a major improvement over SALSA, demonstrate the potential benefit of large training samples of repeated examples.

Table 2: Secret recovery for different Data budgets (DB), and repetition levels, for secrets with Hamming weight $4, 5$ and $6$. –: no secret recovered, ✓: recovery in the noiseless case only, ✓✓: recovery in all cases.

| Data budget | Hamming weight 4 | | | | | Hamming weight 5 | | | | | Hamming weight 6 | | | | |
|---|---|---|---|---|---|---|---|---|---|---|---|---|---|---|---|
| | 1x | 2x | 4x | 10x | 20x | 1x | 2x | 4x | 10x | 20x | 1x | 2x | 4x | 10x | 20x |
| 1M | – | – | – | ✓ | ✓ | – | – | – | – | – | – | – | – | – | – |
| 4M | ✓ | ✓ | ✓✓ | ✓✓ | ✓✓ | – | – | – | ✓ | ✓ | – | – | – | – | – |
| 20M | ✓ | ✓ | ✓✓ | ✓✓ | ✓✓ | – | – | ✓ | ✓ | ✓✓ | – | – | – | ✓ | ✓ |
| 50M | ✓✓ | ✓✓ | ✓✓ | ✓✓ | ✓✓ | – | – | ✓ | ✓✓ | ✓✓ | – | – | ✓ | ✓ | ✓✓ |

We then experiment with **reduced data**, for $n = 256 \log_2 q = 20$ (34 cruel region size, Table 3), and $n = 512 \log_2 q = 28$ (228 cruel region size, Table 4). We consider secrets with $h = 4$ and $5$, for each setting and value of $h$, we sample $4$ different secrets, and train 16 models (with different weight initializations). We considered the experiment is successful if one model recovers all cruel bits, and report the number of secrets recovered out of four.

For $n = 256$, secrets with $4$ cruel bits are recovered, given very large data budgets (400 million different examples). Repetition allows recovery from smaller datasets. Secrets with $h = 5$ can only be recovered with large training sets. For dimension 512, secrets with $4$ cruel bits can be recovered from 20 million of examples, repeated 10 times. Secrets with $5$ cruel bits can be recovered with a training budget of 200 million examples. Models trained on large sets of repeated examples do recover secret with more than 3 cruel bits, a clear improvement over previous ML attacks.

Table 3: $n = 256, \log_2 q = 20$

| Data budget | 4\|5 cruel bits | | | | | |
|---|---|---|---|---|---|---|
| | 1x | 2x | 5x | 10x | 20x | 100x |
| 20M | 0\|0 | 0\|0 | 0\|0 | 0\|0 | 0\|0 | 0\|0 |
| 50M | 0\|0 | 0\|0 | 0\|0 | 0\|0 | 1\|0 | - |
| 100M | 0\|0 | 0\|0 | 1\|0 | 1\|0 | 1\|0 | - |
| 200M | 0\|0 | 1\|0 | 2\|0 | 3\|1 | - | - |
| 400M | 3\|1 | 4\|1 | 4\|1 | - | - | - |

Table 4: $n = 512, \log_2 q = 28$

| Data budget | 4\|5 cruel bits | | | | | |
|---|---|---|---|---|---|---|
| | 1x | 2x | 5x | 10x | 20x | 100x |
| 20M | 0\|0 | 0\|0 | 0\|0 | 1\|0 | 1\|0 | 1\|0 |
| 50M | 0\|0 | 1\|0 | 1\|0 | 1\|0 | 1\|0 | - |
| 100M | 0\|0 | 1\|0 | 1\|0 | 1\|0 | 2\|0 | - |
| 200M | 0\|0 | 2\|0 | 2\|0 | 2\|1 | - | - |
| 400M | 1\|0 | 4\|0 | 4\|1 | - | - | - |

Number of secrets recovered. "-" indicates experiments could not be run: compute budget is too large

These results also shed new light on the role of cool bits. In theory, secrets should be easier to recover with $n = 256, \log_2 q = 20$ than with $n = 512, \log_2 q = 28$: the dimension is smaller, the the BKZ-reduction rate is much better (34 cruel columns vs 224). Yet, with repetition, secrets with $4$ cruel bits can be recovered with 20 million examples only for $n = 512$, vs 50 millions for $n = 256$. Somehow, a higher reduction rate, and more cool bits, seems to complicate the recovery of cruel bits. In the next section, we investigate this counter-intuitive observation.

## 5 TAMING THE COOL BIT NOISE: STEPWISE REGRESSION

ML-based attacks recover secrets by comparing model predictions for close values of $\mathbf{a}$. If $\mathbf{a}' = \mathbf{a} + \frac{q}{2}\mathbf{e}_i$, with $\mathbf{e}_i$ the $i$-th base vector, then the difference between the associated values of $b = \mathbf{a} \cdot \mathbf{s} + \epsilon$

$$b' - b = (\mathbf{a}' - \mathbf{a}) \cdot \mathbf{s} + \epsilon' - \epsilon = \frac{q}{2}\mathbf{s}_i + \epsilon' - \epsilon \mod q$$

has mean zero if $\mathbf{s}_i = 0$, and $q/2$ if $\mathbf{s}_i = 1$. If the transformer produces good predictions of $b$ and $b'$, i.e. $\mathcal{T}(\mathbf{a}) \approx b$ and $\mathcal{T}(\mathbf{a}') \approx b'$ (with $\mathcal{T}$ the transformer prediction), the difference $|\mathcal{T}(\mathbf{a}) - \mathcal{T}(\mathbf{a}')|$, averaged on a sample of vectors $\mathbf{a}$, should allow us to guess the corresponding secret bit.

Previous research (section 5 of (Li et al., 2023a)) suggests that transformers struggle to learn modular dot products when the sum $|\mathbf{a} \cdot \mathbf{s}|$ becomes larger than a multiple of $q$. With reduced data, this can happen in two cases: when the number of (unreduced) cruel bits in the secret exceeds a relatively

small value, **but also** when the size of the reduced, cool, region becomes large. We believe this accounts for the observation, at the end of the previous section, that the cruel bits for the "hard setting" $n = 512 \log_2 q = 28$, were easier to recover, than those of the easier setting $n = 256 \log_2 q = 20$, which enjoyed a higher reduction factor. In this section, we investigate cool bit recovery, especially in the case when BKZ reduction is high, and the secret has a lot of non-zero cool bits.

Previous research assumes that once the cruel bits are known, cool bit recovery is easy, and proposes a linear regression recovery method (Wenger et al., 2025). Once the cruel bits are guessed, the quantity $b_{cool} = b - \mathbf{a}_{cruel} \cdot \mathbf{s}_{cruel} = \mathbf{a}_{cool} \cdot \mathbf{s}_{cool} + \epsilon$ is computed ($\mathbf{s}_{cool/cruel}$ represent the restriction of $\mathbf{s}$ to the cool/cruel coordinates), and linear regression is used to predict $\mathbf{s}_{cool}$ from $\mathbf{a}_{cool}$ and $b_{cool}$.

The use of linear regression, here, is dubious for two reasons. First, we know that the coordinates of $\mathbf{a}$ are not correlated, and that the secret bits are independent (see Appendix B). Linear regression, on the other hand, will compute and invert the test sampled covariance matrix $\mathbf{A}^T \mathbf{A}$, which will have non diagonal elements due to population error that will be amplified by matrix inversion. Second, linear regression ignores the fact that the dot product is computed modulo $q$. As a result, it underestimates the contributions of non-zero bits in the secret (which can "wrap" to zero when their sum exceeds $q$). Summarizing, we believe that cool bits should better be recovered one by one than all at once, and zero bits are easier to recover that non-zero bits.

For this reason, we propose a new method for cool bit recovery, based on stepwise regression (Efroymson, 1960). Once the cruel bits have been guessed, their contribution is subtracted from $b$, and we compute $b_{cool} = b - \mathbf{a}_{cruel} \cdot \mathbf{s}_{cruel}$. As in the previous method, we run a linear regression on remaining cool bits, but look for the feature with the *lowest contribution*, and assign the value zero to the corresponding secret bit. We then remove this bit from $(\mathbf{a}_{cool}, b_{cool})$, and repeat the process until we get the known value of $h$ (or a value in a predefined range, if $h$ is not assumed to be known).

Stepwise regression recovers the zero bits of the secrets. At first, because the secret is sparse, there are more zeroes than ones, but after a number of steps, the remaining bits of the secrets are mostly ones. In that situation, it is more efficient to consider the dual problem: flip all the remaining cool bits, and perform the regression on $b_{dual} = \mathbf{a}_{cool}(\mathbf{1} - \mathbf{s}_{cool}) = \mathbf{a}_{cool}^\top \mathbf{1} - b_{cool}$. We call this variant *dual stepwise regression*: we run stepwise regression until the undiscovered cool bits have more ones than zeroes. Then, we alternate between direct and dual recovery. For instance, if we know, or estimate that, the secret has 10 cool zeros and 5 ones, we will apply the direct step 6 times, before alternating dual and direct. We provide the pseudocode in Appendix E.

Tables 5 and 6 compare linear, stepwise and dual stepwise regression in the two settings with large BKZ reduction (results for the other settings can be found in Appendix F). We consider secrets with 4 to 8 cruel bits, and set the Hamming weight, so that the ratio of cruel bits over $h$ is constant, and equal to $c/n$ for this setting. For each value of $h$ we run models on 20 different secrets, and report the number of secrets recovered (assuming cruel bits are known), for different numbers of LWE examples used for recovery. Overall, stepwise regression outperforms linear regression, and dual stepwise regression achieves the best results.

Table 5: $n = 256, \log_2 q = 20$

| Cool bits | Linear | | Stepwise | | Dual | |
|---|---|---|---|---|---|---|
| (Total $h$) | 2M | 20M | 2M | 20M | 2M | 20M |
| 26 (30) | 2 | 13 | 3 | 20 | 13 | 20 |
| 32 (37) | 0 | 5 | 0 | 20 | 5 | 20 |
| 38 (44) | 0 | 1 | 0 | 11 | 1 | 20 |
| 45 (52) | 0 | 0 | 0 | 3 | 0 | 18 |
| 52 (60) | 0 | 0 | 0 | 1 | 0 | 14 |

Table 6: $n = 512, \log_2 q = 41$

| Cool bits | Linear | | Stepwise | | Dual | |
|---|---|---|---|---|---|---|
| (Total $h$) | 1M | 4M | 1M | 4M | 1M | 4M |
| 40 (44) | 20 | 20 | 20 | 20 | 20 | 20 |
| 50 (55) | 20 | 20 | 20 | 20 | 20 | 20 |
| 60 (66) | 13 | 18 | 17 | 20 | 20 | 20 |
| 70 (77) | 6 | 17 | 15 | 19 | 20 | 20 |
| 80 (88) | 0 | 12 | 10 | 18 | 19 | 20 |

Cool bits recovery out of 20 secrets.

In both settings, dual stepwise regression allows to recover secrets with 8 cruel bits. For $n = 256$, linear regression cannot recover secrets with more than 4 cruel bits with 2 million LWE examples, and 6 with 20 million. Dual stepwise regression recovers 6 with 2 millions, and 8 with 20. For $n = 512$, dual recovery allows for the recovery of 19 out of 20 secrets with 8 cruel bits, with only 1 million LWE examples. This clearly demonstrates the benefits of stepwise regression.

## 6 OVERALL SECRET RECOVERY

In this final section, we present the overall results of our attack, for the four settings. We consider two measures of success. First, we consider the maximum Hamming weight and number of cruel bits that can be recovered, as a function of the training set size and the level of repetition. Then, for a given Hamming weight, we estimate the proportion of all secrets that our method can recover (depending on the number of cruel and cool bits each secret has). In these experiments, we assume that the attacker only knows the public available secret Hamming weight $h$. In particular, the attacker does not know the cruel bits. We generate 15,000 model based attempts of the cruel bits, and use dual stepwise recovery for cool bit recovery.

For $n = 256 \log_2 q = 12$ (Table 7), SALSA attacks recover secrets with $h = 8$, and Cool and cruel with $h = 12$. We recover secrets with $h = 14$, and 5 cruel bits. Our best results are achieved with small data budgets (4 million examples) and large repetition (15 times). Note that models trained on BKZ-reduced data achieve the same results are those trained on synthetic data. For $n = 512$ and $\log_2 q = 28$ (Table 8), the other hard setting (low modulus, larger cruel region), we recover $h = 12$ (and 5 cruel bits), like the cool and cruel attack. Again, performances on reduced and synthetic data are the same, and repetition matters more than the number of reduced examples.

**Highest Hamming weight and cruel bits recovered - binary secret.**

Table 7: $n = 256, \log_2 q = 12$.

| | Repetition | | | | |
|---|---|---|---|---|---|
| | 1x | 2x | 5x | 15x | 50x |
| BKZ-reduced data | | | | | |
| 4M | 10/3 | 10/3 | 12/4 | **14/5** | 14/5 |
| Synthetic data | | | | | |
| 4M | 10/3 | 10/3 | 10/3 | **14/5** | 14/5 |
| 20M | 10/3 | 10/3 | 10/3 | 14/4 | 14/5 |
| 50M | 10/3 | 10/3 | 12/3 | 14/5 | - |
| 100M | 10/4 | 10/4 | 12/5 | - | - |
| 200M | 12/5 | 12/5 | 12/5 | - | - |
| 400M | 12/5 | 12/5 | - | - | - |

Best of 80 models.

Table 8: $n = 512, \log_2 q = 28$

| | Repetition | | | | |
|---|---|---|---|---|---|
| | 1x | 2x | 5x | 15x | 50x |
| BKZ-reduced data | | | | | |
| 4M | 10/3 | 10/3 | 10/4 | 12/4 | 12/4 |
| 20M | 10/4 | 10/4 | 10/5 | **12/5** | 12/5 |
| 50M | 10/4 | 10/4 | 10/3 | 12/5 | - |
| Synthetic data | | | | | |
| 4M | 10/3 | 10/3 | 10/3 | 10/3 | 12/4 |
| 20M | 10/3 | 10/4 | **12/5** | 12/5 | 12/5 |
| 50M | 10/3 | 10/4 | 12/4 | 12/4 | - |
| 100M | 10/3 | 12/4 | 12/5 | - | - |
| 200M | 12/4 | 12/4 | 12/4 | - | - |

Best of 80 models.

Table 9: $n = 256, \log_2 q = 20$

| | Repetition | | | | |
|---|---|---|---|---|---|
| | 1x | 2x | 5x | 15x | 50x |
| BKZ-reduced data | | | | | |
| 4M | 55/6 | 55/6 | 55/7 | 60/8 | 65/8 |
| 20M | 55/6 | 55/6 | 60/7 | 60/8 | 65/8 |
| 50M | 60/8 | 65/8 | 65/8 | 65/8 | - |
| 100M | 65/8 | 65/8 | **70/8** | - | - |
| 200M | 65/8 | 70/8 | 70/8 | - | - |
| 400M | 65/8 | 70/8 | - | - | - |
| Synthetic data | | | | | |
| 4M | 55/6 | 60/7 | 60/7 | 60/7 | 60/8 |
| 20M | 60/8 | 65/8 | 65/8 | 65/8 | 65/8 |
| 50M | 65/7 | 65/8 | **70/8** | 70/8 | - |
| 100M | 65/8 | 65/8 | 70/8 | - | - |
| 200M | 65/8 | 65/8 | 70/8 | - | - |
| 400M | 65/8 | 65/8 | - | - | - |

Best of 80 models.

Table 10: $n = 512, \log_2 q = 41$

| | Repetition | | | | |
|---|---|---|---|---|---|
| | 1x | 2x | 5x | 15x | 50x |
| BKZ-reduced data | | | | | |
| 4M | 70/4 | 70/4 | 70/4 | 70/6 | 70/6 |
| Synthetic data | | | | | |
| 4M | 70/4 | 70/4 | 70/4 | 70/5 | 70/6 |
| 20M | 70/6 | 70/6 | 70/7 | **75/7** | 75/7 |
| 50M | 70/6 | 70/6 | 70/6 | 75/7 | - |
| 100M | 70/6 | 70/6 | 70/6 | - | - |
| 200M | 70/6 | 70/6 | - | - | - |

Best of 80 models.

The improvement over previous works is larger in settings with higher reductions, i.e. smaller cruel region. For $n = 256 \log_2 q = 20$, secrets with $h = 70$ and 8 cruel bits can be recovered, vs $h = 33$

for SALSA methods (Table 9). For $n = 512 \log_2 q = 41$, secrets with $h = 75$ and 7 cruel bits are recovered, vs 63 in SALSA, and 60 in the cool and cruel attack (Table 10). As before, there is little difference between models trained on reduced and synthetic data. In Appendix G we show **similar results on ternary secret**: for $n = 256 \log_2 q = 20$, secrets with $h = 55$ and 8 cruel bits can be recovered compared to $h = 24$ from SALSA pipeline. Similarly, for $n = 512 \log_2 q = 41$, we recover secrets with $h = 75$ and 7 cruel bits vs 66 in SALSA.

The metric used so far, the highest recoverable Hamming weight, can be misleading. A secret with a very large Hamming weight could, with a very low probability, have a every small number of cruel bits, and be recoverable. We therefore consider the proportion of all secrets with a given $h$ that our attack can recover. For any value of $h$ and $k$ the number of cruel bits in the secret, we compute our model recovery rate $r(h, k)$, and the probability $p(h, k)$ that a random secret has $k$ cruel bits, which follows a hyper-geometric distribution. Then, the expected recovery rate is $\mathcal{E}(h) = \sum_{k=0}^{h} p(h, k) r(h, k)$.

**Expected recovery rate for different Hamming weights.**

Table 11: $\mathcal{E}(h)$ **for** $n = 256, \log_2 q = 20$

| $h$ | 33 | 55 | 60 | 65 | 70 |
|---|---|---|---|---|---|
| SALSA | 33% | 4% | 2% | 1% | 0% |
| Ours | **98%** | **71%** | **60%** | **49%** | **38%** |

Table 12: $\mathcal{E}(h)$ **for** $n = 512, \log_2 q = 41$

| $h$ | 63 | 65 | 70 | 75 |
|---|---|---|---|---|
| SALSA | 15% | 14% | 10% | 7% |
| Ours | **91%** | **89%** | **72%** | **63%** |

In Tables 11 and 12, we compare expected recovery rate $\mathcal{E}(h)$ for our method and SALSA. Our method not only recovers higher $h$, but it is much more reliable. For $n = 256$, $\log_2 q = 20$ and $h = 33$, VERDE recovers 33% of secrets (up to 3 cruel bits), while we recover **98%** (up to 8 cruel bits). For $n = 512$, $\log_2 q = 41$ and $h = 63$, VERDE recovers 15% of secrets, we recover **91%**.

## 7 SCALING LAWS

We define model based attempts $A$ as the number of attempts needed to get the correct cruel bits (attempts are generated from the distinguisher output. We choose $A$ as a more fine-grained metric of model performance on secret recovery. For $n = 256 \log_2 q = 20$, and the binary case, we present empirical laws relating the number of model based attempts, $A$, required to recover a secret for model size $N$, data amount $D$, and repetitions $R$ for a fixed secret $s$ with Hamming weight $h$.

**Model parameters law:** To understand any scaling pattern between model size $N$ and model based attempts $A$, we vary the embedding dimension between 256 and 1024 and the number of layers between 4 and 12. We test different model sizes for three different secrets with Hamming weight $h = \{60, 65, 70\}$. We use 100 million data and 1 repetition. As shown in Figure 1, the number of model based attempts is not improved by an increase in model parameters.

**Data-repetition law:** We fix the model embedding dimension at 256 and 4 layers. We vary the distinct data amount from 1 million to 400 millions and data repetitions $R$ from 1 to 50. We define $D$ as the total training data, equal to distinct training data times the data repetition factor $R$. Figure 2 presents the results for the best-performing model out of 8 different initializations. We propose the following functional form: $\ln(A_R) = C_R - \alpha_R \ln(D)$ and we report the fitted parameters using least square errors between $(\ln(D), \ln(A_R))$ across 5 distinct $R$ regimes in Table 13. Notably, our experiments reveal that $\alpha_1$ is considerably lower than $\alpha_R$, $R > 1$. Therefore, data repetition does not just diminish the required number of distinct (costly) samples but actually reshapes the scaling power law of model based attempts $A$ as a function of $D$. Repeated data is crucial for recovering a secret with Hamming weight $h = 70$. **Simply increasing data is important but insufficient**; a careful data strategy is necessary to tackle the LWE problem.

## 8 DISCUSSION AND CONCLUSION

We introduced three main techniques for improving secret recovery in ML attacks: using larger training sets, repeating training examples, and stepwise regression for cool bits recovery. This brings considerable improvement over previous attacks, both in terms of the maximum recoverable

| $R$ | 1 | 2 | 5 | 15 | 50 |
|---|---|---|---|---|---|
| $C_R$ | 26.9 | 35.6 | 38.1 | 42.8 | 51.3 |
| $\alpha_R$ | 0.70 | 1.31 | 1.45 | 1.58 | 1.95 |
| 95% $\alpha_R$ Confidence Interval | 0.61 - 0.79 | 1.25 - 1.4 | 1.38 - 1.51 | 1.44 - 1.71 | 1.76 - 2.26 |

Table 13: Empirical scaling power law fitted parameters. Confidence interval is computed using bootstrapping.

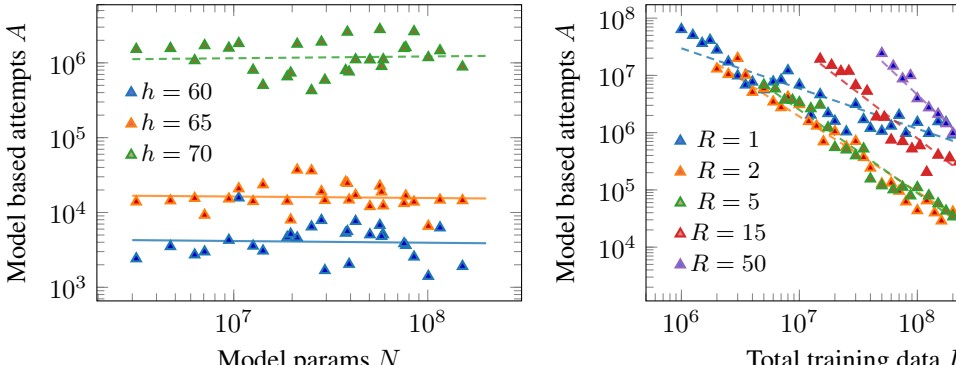

Figure 1: **Model parameters $N$ vs model based attempts $A$ for three secrets with different Hamming weight $h$.**

Figure 2: **Total training data $D$ and repetition $R$ vs model based attempts $A$ for one secret with Hamming weight $h = 70$. Total distinct data can be computed as $D/R$.**

Hamming weight and the proportion of secrets recovered for a given $h$. We additionally presented an empirical scaling power law that relates model based attempts to data amount and data repetitions.

**Limitations.** Our attack targets Learning With Errors in its basic form. We make no claim about its applicability to other PQC algorithms, but believe it can be adapted to the variants of LWE considered in the SALSA papers (Ring and Module LWE, notably). Our experiments focus on sparse binary and ternary secrets, but the attack can be adjusted to small secrets, following the methods described in VERDE (Li et al., 2023a). Finally, we target sparse secrets: with about 5% density for the harder settings, and 14 and 27% in the easier ones.

Our conclusions extend in two directions. First, larger training sets, used for many epochs, are key to recovering the cruel bits. This confirms previous observations about the importance of repeated data (Charton & Kempe, 2024; Saxena et al., 2024). The need for large training sets, and data for stepwise regression, may limit the applicability of our attack: whereas large reduced datasets can be created from small eavesdropped samples (Li et al., 2023b), BKZ reduction is costly, both in terms of time and computing resources. We believe synthetic data can be used to simulate experimental settings, without the costly preprocessing step. This has two key implications: (1) it allows synthetic data to be used for experimentally establishing scaling laws for breaking arbitrary binary, ternary, or small secrets and (2) it suggests that ML attacks could be improved using pretraining on synthetic data, which is essentially free to generate compared to data reduction via lattice reduction techniques.

Our second conclusion is the benefit of stepwise regression. Most statistical handbooks consider stepwise regression a subpar alternative (Smith, 2018), because it does not take into account possible correlations between input features. We believe our conclusions stem from the very specific nature of the data considered here: the columns of matrix **A** are uncorrelated, even after BKZ-reduction, and stepwise regression enforces this inductive bias. In most problems of statistics, on the other hands, features can be correlated, and stepwise regression is harmful.

Our results improve the performance of ML-based attacks on LWE. By investigating scaling laws on LWE for the first time, we provide some insights on which strategies can be used to tackle harder versions of the LWE problem. On a broader level, this line of research is important, because PQC systems like LWE are the future standards for safe digital transactions, and the community must have a clear understanding of their potential limitations and weaknesses, notably those relative to the use of sparse and small secrets, before they are deployed at a large scale. Any weakness discovered now, and taken into account in the standard, is one less future attack against PQC.

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

# APPENDIX

## A  BKZ-REDUCED DATASET

We report some statistics in Table 14 of the BKZ-reduced datasets. $c$ is the size of the cruel region (coordinates with variance larger than half the variance of the uniform law), $\sigma_{cool}$ is the standard deviation of the coordinates of the cool region centered in $[-q/2, q/2]$, $\rho$ is the mean of absolute values of off-diagonal pairwise correlation from the covariance matrix of the reduced samples and $\sigma_\epsilon$ is the standard deviation of $\mathbf{R}\epsilon$ centered in $[-q/2, q/2]$.

It is important to note that reduction creates a cool region of size $n - c$ but introduces an error term with standard deviation approximately equal to $q/\sqrt{12}$ (the standard deviation of one cruel bit).

Table 14: **Reduction statistics**. We report dataset size, CPU hours, $c$ (size of the cruel region), $\sigma_{cool}$ (standard deviation of the cool region), $\rho$ (average non-diagonal pairwise absolute value correlation from the covariance matrix of $\mathbf{RA}$) and $\sigma_\epsilon$ (standard deviation of $\mathbf{R}\epsilon$).

| $n$ | $\log_2 q$ | Dataset size (millions) | CPU hours (millions) | $c$ | $\sigma_{cool}$ | $\rho$ | $\sigma_e$ |
|---|---|---|---|---|---|---|---|
| 256 | 12 | 4 | 0.1 | 143 | $0.30q/\sqrt{12}$ | 0.18% | $0.88q/\sqrt{12}$ |
| 256 | 20 | 400 | 40.4 | 34 | $0.23q/\sqrt{12}$ | 1.05% | $0.90q/\sqrt{12}$ |
| 512 | 28 | 60 | 1.0 | 224 | $0.19q/\sqrt{12}$ | 0.12% | $0.70q/\sqrt{12}$ |
| 512 | 41 | 4 | 0.1 | 46 | $0.15q/\sqrt{12}$ | 0.18% | $0.80q/\sqrt{12}$ |

## B  SYNTHETIC DATASET

Some of the experiments in the manuscript require very large training sets (up to 400 million reduced data), which requires a lot of computing resources. We reduce 400 million LWE samples for $n = 256$ $\log_2 q = 20$, but for the three other settings, we relied on synthetic data to complement smaller BKZ-reduced datasets (see Table 14).

To generate synthetic data, we observe that after reduction the coordinates are uncorrelated (i.e. the off-diagonal terms of the covariance matrix $(\mathbf{RA})^T(\mathbf{RA})$ are very small), i.e. the mean absolute value $\rho$ is close to 0. This suggests a method for generating synthetic data. One million LWE samples are reduced using BKZ, and we use these reduced examples to measure $c$, $\sigma_{cool}$ and $\sigma_\epsilon$. The synthetic reduced $\mathbf{a}$ are then generated by sampling the first $c$ coordinates from a uniform distribution, and the $n - c$ others from a centered distribution with variance $\sigma_{cool}$. We then compute $b = \mathbf{a} \cdot \mathbf{s} + \epsilon$, with $\epsilon$ a centered discrete Gaussian variable, with standard deviation $\sigma_\epsilon$. Note that because all unreduced columns are assumed to have the variance of the uniform distribution, the synthetic data is a little less reduced than the BKZ-reduced data.

When BKZ-reduction is applied, the original matrix $\mathbf{A}$ and vector $\mathbf{b}$ of LWE samples are subjected to a linear transformation $\mathbf{R}$. The columns of $\mathbf{RA}$ are not correlated, and have a block structure: the $c$ first column have high variance, and the $n - c$ last columns low variance. Multiplying $\mathbf{RA}$ by any block diagonal matrix of the form

$$\mathbf{O} = \begin{bmatrix} \Omega_1 & 0 \\ 0 & \Omega_2 \end{bmatrix} \tag{1}$$

with $\Omega_1$ and $\Omega_2$ two $c \times c$ and $(n - c) \times (n - c)$ matrices, near-orthogonal, (i.e. with condition numbers close to 1), should have little impact on the distribution of cool and cruel bits. This means we can generate, from BKZ-reduced samples $\mathbf{RA}$, a lot of synthetic samples, with the same secret, reduction factor, and noise, by sampling quasi-orthogonal matrices $\mathbf{O}$, and generating new data $\mathbf{ORA}$ and $\mathbf{ORb}$. We leave an analysis of this data augmentation method for future work.

## C  COOL AND CRUEL EMBEDDING ABLATION

For $n = 256 \log_2 q = 20$ we train 16 different models with 400 million samples and 2 repetitions for 5 different secrets with 4 cruel bits. In Table 15 we compare the recovery rate (i.e. if the secret is fully recovered) with or without the cool and cruel embedding introduced in Section 3. We highlight that the new embedding allows for recoveries on all secrets.

Table 15: **Cool and cruel embedding comparison**.

| $h$ | CC embedding disabled | CC embedding enabled |
|---|---|---|
| 28 | **5/16** | **8/16** |
| 30 | 0/16 | **4/16** |
| 33 | **3/16** | 7/16 |
| 34 | 0/16 | **5/16** |
| 36 | 0/16 | **1/16** |

## D  BIAS IN ANGULAR EMBEDDINGS AND WAYS TO CIRCUMVENT IT

To predict $b = \mathbf{a} \cdot \mathbf{s} + \epsilon$ from $\mathbf{a}$, our transformer uses an angular embedding. The model outputs a point $P(x, y)$ on the real plane, the integers $i \in \{0, \dots, q-1\}$ are encoded as the points $B_i(\cos(\frac{2\pi i}{q}), \sin(\frac{2\pi i}{q}))$ on the unit circle, and the model prediction is the point $B_i$ closest to $P$. If $P$ has polar coordinates $(r\cos(\theta), r\sin(\theta))$, with $r > 0$, the point $B_{i_0}$ closest to $P$ verifies $i_0 = \operatorname{argmin}_i |\theta - \frac{2\pi i}{q}|$.

During training, the model learns to minimize a Mean Square Error (MSE) loss. If the angular embedding of $B$ is $(x_0, y_0)$, and if the model predicts $P(x, y)$, the loss is $l = (x - x_0)^2 + (y - y_0)^2$. Since the possible values of $b$ are uniformly distributed on the unit circle, we can assume, without loss of generality, that $B = (1, 0)$. Therefore, the loss is

$$l = (1 - r\cos(\theta))^2 + r^2 \sin^2(\theta) = 1 + r^2 - 2r\cos(\theta).$$

During the early stages of training, the model has not learned modular arithmetic and $b$ is predicted at random. Suppose that all model predictions lie at a distance $r$ of the origin, the average MSE loss is the integral of $l$ over all possible angles, so

$$\mathcal{L}(r) = \frac{1}{2\pi r} \int_{-\pi}^{\pi} (1 + r^2 - 2r\cos(\theta)) r \, d\theta = 1 + r^2.$$

Therefore, for a clueless model (a model that predicts $b$ at random), the average loss is $1 + r^2$, and the optimizer can reduce the loss just by making $r$ smaller. Model predictions will therefore tend to collapse towards the origin, at which point the loss becomes constant ($l(0) = 1$ no matter the prediction), and nothing can be learned anymore.

Note that collapse only happens when the model cannot predict $b$. If the model learns to predict $b$ up to some error, i.e. that, assuming $B = (1, 0)$ the predicted value of $\theta$ lies in the interval $[-\varepsilon, \varepsilon]$ for some small $\varepsilon$, the average loss then becomes:

$$\mathcal{L}_\varepsilon(r) = 1 + r^2 - 2r \frac{\sin(\varepsilon)}{\varepsilon} = (r - 1)^2 - 2r \left( \frac{\sin(\varepsilon)}{\varepsilon} - 1 \right).$$

This shows that once the model starts learning, model predictions stop collapsing to the origin. In other words, model collapse only happens at the beginning of training. (Note, we assume here that model predictions are uniformly distributed over $[-\varepsilon, \varepsilon]$, this is a simplification. It can be shown that the same phenomenon appears if the distribution of predictions is unimodal, and centered on 0).

To prevent the model from collapsing in the initial phase of training, we add to the loss a penalty $\mathcal{P}(r) = \alpha r^2 + \frac{\beta}{r^2}$. The average loss (for a random prediction of $b$) then becomes

$$\mathcal{L}(r) = 1 + (1 + \alpha) r^2 + \frac{\beta}{r^2}.$$

It reaches a minimum for $\mathcal{L}'(r) = 2(1 + \alpha)r - 2\beta/r^3 = 0$ or $r^4 = \frac{\beta}{1+\alpha}$.

In Table 16 we experiment with two different settings and we report the hardest recovered secret by the two different settings. The first setting fixes $\beta = 1$ and $\alpha = 0$ to force predictions to remain close to the unit circle in the initial, "clueless", phase of learning. The second setting is inherited from Saxena et al. (Saxena et al., 2024), where authors suggest $\alpha = \beta = 0.1$.

Overall the two settings are similar, with the second setting being more successful or at par on almost all data budgets.

Table 16: **Loss comparison**.

| Data budget | Repetitions | $\alpha = 0, \beta = 1$ | $\alpha = 0.1, \beta = 0.1$ |
|---|---|---|---|
| $N = 256 \log_2 q = 20$ | | | |
| 50M | 15x | 60/8 | 65/8 |
| 100M | 5x | 65/8 | 65/8 |
| 200M | 5x | 65/8 | 65/8 |
| 400M | 2x | 65/8 | **70/8** |
| $N = 512 \log_2 q = 28$ | | | |
| 20M | 15x | 12/4 | **12/5** |
| 50M | 15x | 12/4 | 12/4 |

## E  STEPWISE REGRESSION ALGORITHM

We include the pseudocode to replicate the stepwise regression algorithm.

---

**Algorithm 1:** Linear Secret Backward Reduction

---

**Input:** Matrix $\mathbf{a}_{cool} \in \mathbb{Z}_q^{B \times cool}$, vector $b_{cool} \in \mathbb{Z}_q^{cool}$, int $h_{cool}$, flag `use_dual_algo`
**Output:** Secret guess $g$
$g \leftarrow$ vector of length $cool$ with all entries equal to $0$
$active \leftarrow$ vector of length $cool$ with all entries equal to $1$
$ones \leftarrow h_{cool}$
$zeros \leftarrow cool - h_{cool}$
**while** $(\texttt{use\_dual\_algo} \wedge |active| > 0)$ **or** $(\neg\texttt{use\_dual\_algo} \wedge |active| > h_{cool})$ **do**
  $\quad use\_dual \leftarrow (ones > zeros) \wedge \texttt{use\_dual\_algo}$
  $\quad$ // One-step regression
  $\quad$ **if** $use\_dual$ **then**
  $\qquad \lfloor\ b \leftarrow (\mathbf{a}_{cool}^T \mathbf{1} - b) \bmod q$
  $\quad X \leftarrow \mathbf{a}_{cool}[:, active]$
  $\quad y \leftarrow b$
  $\quad coef \leftarrow \arg\min_c \|y - Xc\|_2^2$
  $\quad coef \leftarrow coef / \max(|coef|)$
  $\quad$ // Remove weakest feature
  $\quad j^* \leftarrow \arg\min_j |coef_j|$
  $\quad active[j^*] \leftarrow 0$
  $\quad$ **if** $use\_dual$ **then**
  $\qquad b \leftarrow (b - A[:, j^*]) \bmod q$
  $\qquad g[j^*] \leftarrow 1$
  $\qquad \lfloor\ ones \leftarrow ones - 1$
  $\quad$ **else**
  $\qquad \lfloor\ zeros \leftarrow zeros - 1$
**if** $\neg\texttt{use\_dual\_algo}$ **then**
  $\quad \lfloor\ g[active] \leftarrow 1$
**return** $g$

---

# F  LINEAR REGRESSION

Similar to Section 5 we compare linear, stepwise and dual stepwise regression and we report the cool bits recovery for the two harder settings, where the BKZ-reduced data has a larger cruel region. As shown in Tables 17, 18, 19 and 20 dual stepwise regression shows the best performance.

Table 17: **Cool bits recovery $n = 256, \log_2 q = 12$ assuming cruel bits have been recovered**

| Cool bits | Total $h$ | Linear regression | | | Stepwise regression | | | Dual stepwise regression | | |
|---|---|---|---|---|---|---|---|---|---|---|
| | | 1M | 2M | 4M | 1M | 2M | 4M | 1M | 2M | 4M |
| 8 | 18 | 20 | 20 | 20 | 20 | 20 | 20 | 20 | 20 | 20 |
| 18 | 41 | 8 | 20 | 20 | 20 | 20 | 20 | 20 | 20 | 20 |
| 23 | 52 | 0 | 3 | 7 | 0 | 9 | 20 | 13 | 20 | 20 |
| 28 | 63 | 0 | 0 | 0 | 0 | 0 | 2 | 0 | 3 | 19 |

Table 18: **Cool bits recovery $n = 256, \log_2 q = 20$ assuming cruel bits have been recovered**

| Cool bits | Total $h$ | Linear regression | | | | Stepwise regression | | | | Dual stepwise regression | | | |
|---|---|---|---|---|---|---|---|---|---|---|---|---|---|
| | | 2M | 4M | 10M | 20M | 2M | 4M | 10M | 20M | 2M | 4M | 10M | 20M |
| 20 | 23 | 5 | 14 | 20 | 20 | 5 | 20 | 20 | 20 | 13 | 20 | 20 | 20 |
| 26 | 30 | 2 | 5 | 11 | 13 | 3 | 10 | 19 | 20 | 13 | 17 | 20 | 20 |
| 32 | 37 | 0 | 2 | 4 | 5 | 0 | 4 | 13 | 20 | 5 | 9 | 20 | 20 |
| 38 | 44 | 0 | 0 | 0 | 1 | 0 | 0 | 3 | 11 | 1 | 7 | 13 | 20 |
| 45 | 52 | 0 | 0 | 0 | 0 | 0 | 0 | 0 | 3 | 0 | 2 | 8 | 18 |
| 52 | 60 | 0 | 0 | 0 | 0 | 0 | 0 | 0 | 1 | 0 | 0 | 1 | 14 |

Table 19: **Cool bits recovery $n = 512, \log_2 q = 28$ assuming cruel bits have been recovered**

| Cool bits | Total $h$ | Linear regression | | | Stepwise regression | | | Dual stepwise regression | | |
|---|---|---|---|---|---|---|---|---|---|---|---|
| | | 1M | 2M | 4M | 1M | 2M | 4M | 1M | 2M | 4M |
| 10 | 18 | 20 | 20 | 20 | 20 | 20 | 20 | 20 | 20 | 20 |
| 30 | 53 | 20 | 20 | 20 | 20 | 20 | 20 | 20 | 20 | 20 |
| 40 | 71 | 15 | 20 | 20 | 20 | 20 | 20 | 20 | 20 | 20 |
| 50 | 89 | 5 | 8 | 13 | 14 | 17 | 18 | 17 | 20 | 20 |

Table 20: **Cool bits recovery $n = 512, \log_2 q = 41$ assuming cruel bits have been recovered**

| Cool bits | Total $h$ | Linear regression | | | Stepwise regression | | | Dual stepwise regression | | |
|---|---|---|---|---|---|---|---|---|---|---|---|
| | | 1M | 2M | 4M | 1M | 2M | 4M | 1M | 2M | 4M |
| 40 | 44 | 20 | 20 | 20 | 20 | 20 | 20 | 20 | 20 | 20 |
| 50 | 55 | 20 | 20 | 20 | 20 | 20 | 20 | 20 | 20 | 20 |
| 60 | 66 | 13 | 15 | 18 | 17 | 20 | 20 | 20 | 20 | 20 |
| 70 | 77 | 6 | 12 | 17 | 15 | 18 | 19 | 20 | 20 | 20 |
| 80 | 88 | 0 | 9 | 12 | 10 | 11 | 18 | 19 | 20 | 20 |

# G  TERNARY SECRETS

We report the hardest recovered ternary secret, based on Hamming weight.

**Highest Hamming weight and cruel bits recovered - ternary secret.**

Table 21: $n = 256, \log_2 q = 12.$

|  | Repetition | | | | |
| --- | --- | --- | --- | --- | --- |
|  | 1x | 2x | 5x | 15x | 50x |
| BKZ-reduced data | | | | | |
| 4M | 10/3 | 10/3 | 10/3 | 10/3 | 10/4 |
| Synthetic data | | | | | |
| 4M | 10/3 | 10/3 | 10/3 | 10/3 | 10/3 |
| 50M | 10/4 | 10/4 | 10/4 | 10/4 | - |
| 200M | 10/4 | 10/4 | **12/5** | - | - |
| 400M | 10/4 | 10/4 | - | - | - |
| Best of 80 models. | | | | | |

Table 22: $n = 512, \log_2 q = 28$

|  | Repetition | | | | |
| --- | --- | --- | --- | --- | --- |
|  | 1x | 2x | 5x | 15x | 50x |
| BKZ-reduced data | | | | | |
| 4M | 8/3 | 8/3 | 8/4 | 8/4 | 8/4 |
| 50M | 8/4 | 10/3 | 10/3 | **10/4** | - |
| Synthetic data | | | | | |
| 4M | 8/3 | 8/3 | 8/3 | 8/3 | 8/4 |
| 50M | 8/4 | 8/3 | 10/3 | 10/3 | - |
| 200M | 10/3 | 10/3 | **10/4** | - | - |
| Best of 80 models. | | | | | |

Table 23: $n = 256, \log_2 q = 20$

|  | Repetition | | | | |
| --- | --- | --- | --- | --- | --- |
|  | 1x | 2x | 5x | 15x | 50x |
| BKZ-reduced data | | | | | |
| 4M | 40/5 | 40/5 | 45/5 | 45/5 | 45/5 |
| 50M | 40/5 | 45/6 | 45/6 | 45/6 | - |
| 200M | 50/6 | **55/8** | 55/8 | - | - |
| 400M | 55/7 | 55/8 | - | - | - |
| Synthetic data | | | | | |
| 4M | 40/5 | 40/5 | 40/5 | 45/5 | 45/5 |
| 50M | 45/5 | 50/6 | 50/6 | 50/6 | - |
| 200M | 45/5 | 50/6 | 50/6 | - | - |
| 400M | 50/6 | 50/7 | - | - | - |
| Best of 80 models. | | | | | |

Table 24: $n = 512, \log_2 q = 41$

|  | Repetition | | | | |
| --- | --- | --- | --- | --- | --- |
|  | 1x | 2x | 5x | 15x | 50x |
| BKZ-reduced data | | | | | |
| 4M | 60/5 | 60/5 | 60/5 | 65/5 | 65/5 |
| Synthetic data | | | | | |
| 4M | 60/5 | 60/5 | 60/5 | 65/5 | 65/5 |
| 50M | 65/5 | 70/6 | 70/6 | **75/7** | - |
| 200M | 70/6 | 70/6 | - | - | - |
| Best of 80 models. | | | | | |

