# OpenReview forum: "Improving ML attacks on LWE with data repetition and stepwise regression"
_ICLR.cc/2026/Conference — Submitted to ICLR 2026_

### Official Review · Reviewer_ZUoz · 2025-10-29

**Soundness:** 3
**Presentation:** 3
**Contribution:** 2
**Rating:** 6
**Confidence:** 3

**Summary:**

The paper studies machine-learning-based attacks on the Learning With Errors (LWE) problem using transformer architectures. It introduces two empirical improvements: data repetition and stepwise regression. The authors find that data repetition follows a power-law scaling (data-repetition law) showing that repeating samples accelerates learning and increases recovery accuracy. Stepwise regression slightly improves the recovery of ternary and binary secrets, though it assumes some knowledge of the difficult bits during training.

**Strengths:**

The paper is a carefully executed experimental study: it confirms and extends known heuristics (from modular arithmetic tasks) to a cryptanalytic setting. Its key contribution is to quantify and generalize how repeated exposure to the same samples can significantly improve transformer-based attacks, revealing a clear scaling relationship that had not been explicitly measured in prior work.

**Weaknesses:**

While the paper presents a careful empirical study and introduces useful combinations of known techniques (e.g., data repetition and stepwise regression), the experimental improvements reported, though measurable, appear incremental relative to prior results.

**Questions:**

Q1: The claim that “LWE is provably hard for large n and not too large q” (line 39) is not supported by a citation, and its intended reference is unclear. It would be helpful if the authors clarified what specific result or prior work they refer to.

Q2: It is unclear how $R\Lambda$ (obtained from BKZ reduction, line 146) relates to the operator R later applied to transform (A, b) into (R A, R b). The authors should clarify the intended dimensions of $\Lambda$, whether $R\Lambda$ is the full reduction matrix or a submatrix, and how R is derived from it to ensure that the transformations are well-defined.

Q3: The paper reports a clear empirical improvement from data repetition and formulates a corresponding scaling law (line 420), but the mechanism driving this effect remains vague. The results suggest that repetition possibly increases the effective signal-to-noise ratio of gradient updates etc. I encourage the authors to clarify or justify this interpretation, either by referencing prior analyses or by providing an explicit discussion of how repetition modifies the learning dynamics in their setting.

Q4: The description of the stepwise regression procedure assumes that the cruel bits of the secret can be recovered, yet it is unclear how these bits are actually obtained or estimated in practice. The paper appears to train the regressor under full supervision, which does not correspond to an actual recovery setting. The authors should clarify whether the cruel bits are recovered iteratively (e.g., by conditioning on already predicted bits), through separate statistical inference, or simply assumed known during training. Making this distinction explicit is essential for evaluating the practical validity of the proposed stepwise regression improvement.

---

> ### Author Response · Authors · 2025-11-21
>
> We thank you for your thoughtful feedback.
>
> **Incremental gains**
>
> *[...] the experimental improvements reported, though measurable, appear incremental relative to prior results*
>
> We believe that the improvements are not incremental. In particular,
>
> - The absolute Hamming weight is increased, especially in the $n=256$, $\log q=20$ and $n=512$, $\log q=41$ settings. In the former case, we recover much harder secrets, improving from $h=33$ to $h=70$, hence the secrets we are able to recover with our pipeline are **not sparse anymore**. In the latter, we go from $h=63$ to $h=75$.
> - The expected recovery rate, reported in Table 11 and 12, increases dramatically. In particular, previous work only reported the best results across different settings, without clarifying how many cruel and cool bits were recovered. Previous work showed that ML attacks cannot recover secrets with more than 3 active bits in the cruel region [https://eprint.iacr.org/2024/1229.pdf]. In our case, we can consistently recover a higher number of cruel bits, up to 8.
> - We don’t just scale data, we also show a change in the scaling law once data repetition is employed. This can open the door for further investigations across the cryptographs community.
>
> **Claim on LWE**
>
> *[...] It would be helpful if the authors clarified what specific result or prior work they refer to.*
>
> Thank you for pointing out the missing reference here. We will add the reference to the original security reduction papers by Regev (quantum reduction) and Peikert (http://portal.acm.org/citation.cfm?id=1536414.1536461) and a little more explanation around the assumptions.
>
> **Operator $\Lambda$**
>
> *The authors should clarify the intended dimensions of $\Lambda$, whether $R \Lambda$ is the full reduction matrix or a submatrix, and how R is derived from it to ensure that the transformations are well-defined.*
>
> After sampling $m$ vectors from $\textbf a$, we perform lattice reduction on the $(m + n) \times (m + n)$ matrix $\mathbf \Lambda$ (defined in §3), $\mathbf{\Lambda} =
> \left[\begin{smallmatrix}
> 0 & q\cdot \mathbf{I}_n \\\\
> \omega\cdot \mathbf{I}_m & \mathbf{A}
> \end{smallmatrix}\right]$
>
> Since the BKZ reduction is a linear transformation, it can be represented with linear operators $\[\mathbf C \ \ \mathbf R\]$ such that the norms of $\[\mathbf C \ \ \mathbf R\] \mathbf \Lambda = \[\omega \cdot \mathbf R \ \ \mathbf R \mathbf A + q \cdot \mathbf C\]$ are small. We will add more details in the manuscript.
>
> **Role of repetitions**
>
> *I encourage the authors to clarify or justify this interpretation, either by referencing prior analyses or by providing an explicit discussion of how repetition modifies the learning dynamics in their setting.*
>
> In the original paper [https://arxiv.org/pdf/2410.07041], authors showed that data repetitions, i.e. smaller unique data budgets with more frequent repetition, yields faster learning and better evaluation performance. We validate the claim in our setting where we showed that, at a fixed total number of training examples, fewer attempts are needed to recover the secret when evaluating on unseen $(RA, Rb)$ pairs. We notice that in some settings, training “emerges” only for mild data repetitions, while models do not recover the secrets with no repetitions or too many repetitions.
>
> At a high level, there now exists a strong collection of empirical results in the community, showing that mild repetition works in synthetic (i.e. non-language) datasets, which do not feature redundancy and deduplication. See [https://arxiv.org/pdf/2410.07041] or [https://arxiv.org/pdf/2309.14316] for some of these results. The community doesn’t have theory-backed results yet to explain the empirical results.
>
> **Cruel bits assumptions**
>
> *The authors should clarify whether the cruel bits are recovered iteratively (e.g., by conditioning on already predicted bits), through separate statistical inference, or simply assumed known during training*
>
> Thanks for this question so we can better explain the entire stack. We do not assume supervision on the cruel bits. On each epoch, the full pipeline (end of §3, “Secret recovery”) is the following:
> - Train the transformer on reduced pairs.
> - Run the VERDE distinguisher on 1,000 reduced pairs to rank the cruel coordinates by likelihood of being non‑zero.
> - Enumerate 15,000 candidate cruel‑bit patterns by sampling from this ranking.
> - For each candidate, estimate cool bits with dual stepwise regression and validate the full secret guess using the standard PICANTE statistical test.
>
> Therefore the transformer is responsible for correctly guessing the cruel bits and dual stepwise regression is responsible for correctly guessing the cool bits.
>
> For the purpose of comparing cool‑bit recovery methods, in Section §5 we assume the true cruel bits are known. That’s what Tables 5 - 6 (and 17 - 20) measure: linear vs stepwise vs dual stepwise given the cruel bits. Main results never assume the cruel bits during training.

---

### Official Review · Reviewer_xzek · 2025-10-31

**Soundness:** 3
**Presentation:** 4
**Contribution:** 3
**Rating:** 8
**Confidence:** 2

**Summary:**

Note I have no prior experience in the field of ML attacks in cryptography (or really anything to do with cryptography) so this review is really just an educated guess.

The paper presents a method to attack LWE, an important post-quantum cryptography mechanism. It demonstrates significant improvements over several recent state-of-the-art benchmarks, recovering secrets with greater Hamming weights and more "cruel bits" than previously possible, with greater reliability. The authors also empirically demonstrate a power-law relationship showing how the number of "model based attempts" needed to find a secret scales with the amount of training data and the level of data repetition.

**Strengths:**

The paper appears to be very well motivated and clearly written. It presents a method to attack LWE, an important post-quantum cryptography mechanism. It demonstrates significant improvements over several recent state-of-the-art benchmarks, recovering secrets with greater Hamming weights and more "cruel bits" than previously possible, and with greater reliability. Its originality comes from combining disparate techniques: data repetition and stepwise regression. Although I am not an expert in this field, it would appear that this is a significant result. The synthetic data finding suggests that future attacks could be made much more efficient by first pre-training models on massive, inexpensive synthetic datasets.

**Weaknesses:**

(I am not really qualified to find serious weaknesses in this paper. I used Gemini 2.5 Pro to suggest areas for improvement, a condensed version of which is below. It accords with my understanding of the paper, and does not appear to be disqualifying. If the authors make revisions based on this review, I would suggest considering these points.)

While the paper presents strong results, there are a few areas that could be tightened. The "synthetic data" contribution feels a bit overstated; it's not truly "free" since it requires parameters derived from the costly BKZ reduction in the first place . It's better described as a powerful data amplification technique. Similarly, the "scaling law" analysis is quite narrow, as it's based on just one LWE setting and, for the key repetition finding, a single secret. This makes it hard to generalize as a "law" for LWE. Finally, the new stepwise regression algorithm seems to require knowing the exact number of non-zero "cool bits" ($h_\text{cool}$), which an attacker wouldn't have. The paper would be more convincing if it discussed the impact of estimating this value and how sensitive the attack is to getting that estimate wrong.

**Questions:**

Do you find any of the "weaknesses" above to be unfair? For those that are legitimate, can you propose changes to address them?

---

> ### Author Response · Authors · 2025-11-21
>
> We thank you for your thoughtful feedback.
>
> **Synthetic data**
>
> *The "synthetic data" contribution feels a bit overstated; it's not truly "free" since it requires parameters derived from the costly BKZ reduction in the first place . It's better described as a powerful data amplification technique*
>
> We agree that generating synthetic data is not entirely “free”. However, in our experiments we used only 1M BKZ-reduced samples to generate 400M synthetic data. Producing those 1M reduced data and the subsequent 400M synthetic samples costs roughly 0.1M CPU hours in total. In contrast, generating 400M reduced data directly would require 40.4M CPU hours [see Table 14]. Therefore, using synthetic data would speed up the attacks by 400x.
>
> **Scaling law**
>
> *Similarly, the "scaling law" analysis is quite narrow [...]*
>
> We acknowledge that the study is not comprehensive, and we leave broader exploration (more secrets and additional LWE settings) to future work. For context, generating Figure 1 required ~80,000 V100 GPU‑hours; Figure 2 required ~400,000 V100 GPU‑hours and 2 months of engineering work. We observed similar scaling behaviour for other settings (i.e. $n=512$, $\log q=41$, $h=75$ or $n=256$, $\log q=20$, $h=65$). In these cases, we only spent a fraction of the GPU hours needed to produce something as precise as Figure 2, so we chose not to include these partial results.
>
> **Value of $h_{cool}$**
>
> *Finally, the new stepwise regression algorithm seems to require knowing the exact number of non-zero "cool bits" ($h_{cool}$), which an attacker wouldn't have*
>
> On Page 6, we briefly mentioned that we can even estimate the value of $h_{cool}$ without degrading the dual stepwise regression attack. We will revise the manuscript to make this point clearer.
> Also, in practice, papers on Cryptographic schemes and their applications often give the exact value of $h$ (Hamming Weight).  Given $h$, standard probabilities will tell you the most likely possible numbers of cool bits.

---

### Official Review · Reviewer_eY9Z · 2025-11-01

**Soundness:** 3
**Presentation:** 2
**Contribution:** 2
**Rating:** 4
**Confidence:** 1

**Summary:**

This paper proposes an improved method for uncovering secrets under the Learning With Errors (LWE) problem. It proposes to train on more data and use a revised model architecture to improve the success rate of recovering the secret. On evaluating on larger, more challenging LWE instances, the proposed method demonstrates significant improvements over prior approaches.

**Strengths:**

- Targets an important yet challenging problem of recovering secrets under LWE.
- Superior empirical results to prior methods

**Weaknesses:**

- The technical novelty appears limited and incremental compared to prior work.
- Limited theoretical analysis of why the proposed method works better.
- The presentation could be improved for easier understanding.
- Topic fit into ICLR seems not very strong

**Questions:**

I am not familiar with using machine learning to attack LWE. From what I understand, this paper proposes a set of engineering techniques to improve the empirical performance of recovering secrets under LWE. These techniques include more training data and slightly revised model architecture. However, these technical contributions seem incremental compared to prior work.

What are the key insights that led to these improvements? This paper is written in a way that is hard to follow, especially for readers not already familiar with prior work. Could you clarify, in simple terms, what are the main reasons that the proposed method outperforms prior approaches?

The evaluations show significant empirical improvements, which is encouraging; however, there is limited theoretical analysis of why the proposed method works better. Would it be possible to provide some theoretical insights or intuitions behind the empirical success?

I am concerned about the fit of this paper's topic into ICLR. From the machine learning perspective, improved data sizes and model tweaks seem like engineering optimizations. While the problem of recovering secrets under LWE is a hard and important problem, it is not clear how readers interested in machine learning would benefit from this work. I would appreciate it if there could be more discussion on this.

---

> ### Author Response · Authors · 2025-11-21
>
> We thank you for your thoughtful feedback.
>
> **Novelty**
>
> *[...] These techniques include more training data and slightly revised model architecture. However, these technical contributions seem incremental compared to prior work.*
>
> We believe that we made 3 non-incremental technical contributions, namely:
>
> - We break the long-standing assumption that ML attacks can only recover secrets with at most 3 cruel bits (see https://eprint.iacr.org/2024/1229.pdf). This limitation has been the main bottleneck preventing AI-based attacks from scaling to more difficult secrets.
>
> - We introduce a new decoder for cool bits (dual stepwise regression). In the two difficult settings ($n=256$, $\log q=20$ and $n=512$, $\log q=41$), recovering the cool region was previously the main obstacle preventing secret recovery from scaling further.
>
> - We propose a scaling law in a controlled setting showing that data repetition genuinely accelerates secret recovery. To our knowledge, no previous work has reported a repetition-sensitive scaling law for algorithmic tasks.
>
> **Theoretical analysis**
>
> *Limited theoretical analysis of why the proposed method works better*
>
> We acknowledge that our manuscript is primarily empirical. Nevertheless, we provide explanations for the following components:
>
> - Why stepwise regression succeeds
> - How data repetition helps
> - Why cool bits dominate the learning difficulty on two studied settings
>
> We can further clarify these arguments in the revision.
>
> **Insights**
>
> *Could you clarify, in simple terms, what are the main reasons that the proposed method outperforms prior approaches?*
>
> Below we explain in simple terms why the proposed method is superior to AI and non-AI based attacks.
>
> After data reduction, the vector $a$ is split into two parts: a “cruel” component of size $c$ and a “cool” component of size $n-c$. The cool component has smaller variance than the cruel one.
>
> Previous work showed that ML attacks cannot recover secrets with more than 3 active bits in the cruel region [https://eprint.iacr.org/2024/1229.pdf]. Data repetition enables recovery of more cruel bits (Table 2). This phenomenon is consistent with https://arxiv.org/pdf/2410.07041 which demonstrates that repeating data, i.e. smaller unique data budgets with more frequent repetition, yields faster learning and better evaluation performance.
>
> Once cruel bits are recovered, cool bits are expected to be easier to recover. However, this is not true when the cool region ($n-c$) becomes large, which happens with strong data reduction. In such cases, dual stepwise regression improves substantially over the existing linear regression method. Our method outperforms linear regression because standard linear regression relies on feature correlations in the $A^TA$ matrix that, in this setting, do not exist.
>
> We can add a practical primer for readers interested in this topic and new to AI attacks on LWE.
>
> **Topic fit into ICLR**
>
> *I am concerned about the fit of this paper's topic into ICLR. From the machine learning perspective, improved data sizes and model tweaks seem like engineering optimizations.*
>
> The paper is fundamentally about how deep models learn modular computations, a long‑standing challenge. We provide a data-centric solution (which can be applied across different settings, see [https://arxiv.org/pdf/2410.07041]) with a quantitative scaling law, an optimization remedy with angular anti collapse penalty (which can be applied across other settings, see [https://arxiv.org/pdf/2410.03569]) and hybrid neural+statistical decoder (dual stepwise) that converts weak global signals from the model into exact prediction. These are general ML ideas applicable to general AI audiences.
>
> Additionally, our work reflects a broader trend in AI for mathematics: using AI models to explore open problems in underexplored mathematical domains, including cryptography. ICLR has actively encouraged this type of interdisciplinary work, particularly because LLMs struggle with open problems. Examples include [https://arxiv.org/pdf/1912.01412], [https://arxiv.org/pdf/2006.06462] and [https://arxiv.org/pdf/2308.15594].

---

> > ### Comment · Reviewer_eY9Z · 2025-11-24
> >
> > Thank you for the detailed clarifications. I am not able to determine whether the claimed technical novelty is significant enough, as I lack background knowledge in this topic.
> >
> > I still feel that this paper is more suitable for publication in security venues.
> >
> > As a general reader who lacks domain knowledge of advanced cryptography, I feel that the insights I obtain from this work are rather limited.
> >
> > The following paper is highly related to the topic in this paper, and was published in IEEE Security&Privacy:
> >
> > [Benchmarking Attacks on Learning with Errors](https://ieeexplore.ieee.org/document/11023470)
> >
> > The listed paper also investigates attacks towards LWE, which shares a similar topic to this paper.
> >
> > Could you explain why this paper is better suited for the AI community, rather than the security or cryptography communities?

---

> > > ### Author Response · Authors · 2025-11-24
> > >
> > > Thank you for your response.
> > >
> > > Studying AI for hard computational math problems is intrinsically interesting and important for an audience of researchers in AI: there is so much focus in the community on LLMs, scaling them, adding reasoning capabilities, etc. But the benchmarks they are tested on are mostly math problems. We find radically different situations than what happens with LLMs. For example, we don't have more success by scaling to deeper models with more layers. Thus, finding other techniques such as data strategies, for example exploring data repetition, is very important and foundational for this field. Additionally, we find our discoveries are not tied only to the particular LWE problem, but they can be reused by other researchers: data repetition, better decoding method and scaling law study. We would be happy to add a few sentences sharing our perspective on this motivation into the paper.
> > >
> > > Multiple papers on solving open mathematical problems with AI have been published at AI conferences.  On this specific LWE problem, we found at least 3 papers shared with the entire AI community: [https://proceedings.neurips.cc/paper_files/paper/2022/hash/e28b3369186459f57c94a9ec9137fac9-Abstract-Conference.html], [https://proceedings.neurips.cc/paper_files/paper/2023/hash/a75db7d2ee1e4bee8fb819979b0a6cad-Abstract-Conference.html] and [https://openreview.net/forum?id=le8hVvWi6Q]. On using AI on open hard math problems, there are multiple instances published in AI conference, not in specific math conferences: [https://www.nature.com/articles/s41586-021-04086-x], [https://proceedings.mlr.press/v145/douglas22a/douglas22a.pdf], [https://proceedings.neurips.cc/paper_files/paper/2024/hash/3a1de90699eec7d7f42c91d81f94af16-Abstract-Conference.html].

---

### Official Review · Reviewer_1oft · 2025-11-03

**Soundness:** 3
**Presentation:** 2
**Contribution:** 2
**Rating:** 4
**Confidence:** 2

**Summary:**

The paper studies machine learning (ML) attacks on Learning With Errors (LWE) problems, specifically on instances with **binary or ternary sparse secrets**. The authors revisit prior ML-based attacks that struggled to recover more than a few “cruel bits” (hard-to-learn secret entries) and propose two key modifications:
- (1) training on **very large datasets with repetitions**, and
- (2) replacing standard regression-based recovery with a **stepwise and dual-stepwise regression scheme**.

They test their method across several LWE configurations $(n=256,512, log_2 q=12,20,28,41)$ and report substantial improvements. For instance, they recover up to **8 cruel bits** and **Hamming weights up to 70–75**, exceeding prior ML attacks such as SALSA, PICANTE, and the “Cool & Cruel Bits method”. They also empirically observe a scaling relation
$\ln(A_R) = C_R - \alpha_R \ln D$ between the number of model-based attempts $A_R$ and the amount of distinct data $D$,
with the exponent $\alpha_R$ increasing as repetition $R$ grows.
The paper also describes a synthetic-data variant, designed to match the variance of BKZ-reduced samples,
which reportedly reproduces the BKZ results while avoiding the ~42 million CPU-hours otherwise required.

While I am generally familiar with machine learning methods, I am not an expert in lattice-based cryptography or ML-based cryptanalysis. My assessment focuses on the clarity, methodological soundness, and empirical evidence presented in the main paper.

**Strengths:**

1. **Clear empirical improvement.** The paper demonstrates that a combination of large-scale data with repetition and the proposed stepwise recovery substantially extends the regime where ML attacks succeed. The jump from 2–3 to 7–8 recoverable cruel bits is a convincing improvement over earlier work.

2. **Good experimental depth.** The authors evaluate on several LWE parameter sets, across both binary and ternary secrets, and document results systematically. The tables are detailed and allow for meaningful comparison to existing attacks.

3. **Interesting finding on repetition.** The role of data repetition is well-isolated and carefully quantified. The empirical scaling law with varying repetition levels is a nice insight into how data reuse can amplify model performance rather than simply causing overfitting.

4. **Practical recovery pipeline.** The proposed (dual) stepwise regression for cool-bit recovery is simple and intuitive.
It replaces the regression stage from 'Cool & Cruel' and appears to contribute to the improved results, although direct ablations are provided only in the appendix.

5. **Synthetic data analysis.** Demonstrating that synthetic data can serve as a stand-in for BKZ-reduced data is valuable from a reproducibility perspective. It makes the method much more accessible for future researchers.

**Weaknesses:**

1. **Scope of the scaling law.** While Figure 1 shows results for three secrets with different Hamming weights,
the scaling relation $ln(A_R)=C_R-\alpha_R \ln D$ illustrated in Figure 2 is evaluated only for a single secret ($h = 70$).
The main text does not report multi-secret fits or confidence intervals for the scaling parameters,
which would strengthen the claim.

2. **Comparison on equal compute budget.** The work demonstrates improved recovery but does not normalize for total compute (GPU time + BKZ cost) against prior methods. Some efficiency comparison would help contextualize the gains.

3. **Repetition vs. memorization.** While repetition improves recovery, it remains unclear whether the gain comes from learning more meaningful correlations or from repeated exposure to the same examples.

4. **Stepwise regression sensitivity.** The paper describes switching the regression target once the remaining fraction of 1-entries exceeds 0.5, but does not discuss how this procedure behaves if the true sparsity pattern deviates from that threshold.
Analyzing robustness or providing an adaptive stopping criterion would clarify the method’s reliability.

5. **Lack of contextualization.** The paper does not discuss when the demonstrated recovery capabilities (e.g., number of cruel bits or Hamming weights achieved) would translate into practical cryptographic impact. A brief impact statement or discussion of real-world relevance would help readers interpret the results.

**Questions:**

1. I found the scaling-law section interesting, but it was a bit hard to judge how general it is.
   Could you comment on whether this behavior also appears for other secrets or Hamming weights beyond the example in Figure 2?

2. The role of repetition seems quite central to your results.  Do you have any intuition for why repetition helps so much and where precisely it helps the most?

3. For the stepwise and dual-stepwise regression, it would help me to understand how sensitive the method is to different sparsity levels.  Did you observe cases where it becomes less reliable?

4. Did you try any other regression variants (like ridge or LASSO) before settling on the stepwise approach, or is that left for future work?

5. From a broader perspective, how do your results translate to parameter choices in deployed LWE schemes? Do they suggest specific ranges that might be unsafe under these attacks?

6. At what concrete success threshold (e.g., number of recovered cruel bits, Hamming weight, or % of secret recovered) would you consider this attack a practical threat to an LWE-based scheme in deployment? It would also help if the authors could provide a short *impact statement* that puts the empirical results into context.

---

> ### Author Response · Authors · 2025-11-21
>
> We thank you for your thoughtful feedback.
>
> **Scaling law**
>
> Firstly, we thank you for finding the scaling law work interesting.
>
> *While Figure 1 shows results for three secrets with different Hamming weights, the scaling relation $\ln(A_R) = C_R - \alpha_R \ln D$ illustrated in Figure 2 is evaluated only for a single secret ($h=70$). [...] Could you comment on whether this behavior also appears for other secrets or Hamming weights beyond the example in Figure 2?*
>
> Figure 1 required ~80,000 V100 GPU‑hours; Figure 2 required ~400,000 V100 GPU‑hours and 2 months of engineering work. We observed similar scaling behavior for other settings (i.e. $n=512$, $\log q=41$, $h=75$ or $n=256$, $\log q=20$, $h=65$). In these cases, we only spent a fraction of the GPU hours needed to produce something as precise as Figure 2, so we chose not to include these partial results. We acknowledge that this study is not comprehensive, and we leave to future work expanding to more secrets and additional settings.
>
> *The main text does not report multi-secret fits or confidence intervals for the scaling parameters, which would strengthen the claim.*
>
> We already report the 95% confidence interval for $\alpha_R$ which captures how the number of attempts scales as we increase the data. We will add the 95% confidence interval for $C_R$.
>
> **Comparison on equal compute budget**
>
> *The work demonstrates improved recovery but does not normalize for total compute (GPU time + BKZ cost) against prior methods.*
>
> We followed the same training and evaluation protocol as the SALSA papers to make compute time directly comparable. The pipeline changes do not increase GPU time or BKZ cost. It’s hard to compare against non‑AI attacks, because non-AI attacks don’t succeed in high dimensions with small modulus, so you can only compare to *projected success of non-AI attacks based on heuristics which are known to be buggy*. See the LWE Benchmarking website [https://facebookresearch.github.io/LWE-benchmarking/] for apples-to-apples comparisons across AI and non-AI attacks.
> Nonetheless, we emphasise that our attack recovers secrets with $h=75$, $n=512$ and $\log q=41$. This would take $10^{91}$ brute-force attempts, far beyond any non‑AI attack capabilities.
>
> **Role of repetitions**
>
> *While repetition improves recovery, it remains unclear whether the gain comes from learning more meaningful correlations or from repeated exposure to the same examples. [...] Do you have any intuition for why repetition helps so much and where precisely it helps the most?*
>
> In the original paper [https://arxiv.org/pdf/2410.07041], authors showed that data repetitions, i.e. smaller unique data budgets with more frequent repetition, yields faster learning and better evaluation performance. We validate the claim in our setting where we showed that, at a fixed total number of training examples, fewer attempts are needed to recover the secret when evaluating on unseen $(RA, Rb)$ pairs. We notice that in some settings, training “emerges” only for mild data repetitions, while models do not recover the secrets with no repetitions or too many repetitions.
>
> At a high level, there now exists a strong collection of empirical results in the community, showing that mild repetition works in synthetic (i.e. non-language) datasets, which do not feature redundancy and deduplication. See [https://arxiv.org/pdf/2410.07041] or [https://arxiv.org/pdf/2309.14316] for some of these results. The community doesn’t have theory-backed results yet to explain the empirical results.

---

> ### Author Response · Authors · 2025-11-21
>
> **Stepwise regression**
>
> *Analyzing robustness or providing an adaptive stopping criterion would clarify the method’s reliability.*
>
> In Tables 5–6 and Appendix F we empirically show that dual stepwise regression is robust compared to the other presented variants. We further tried a variant that switches between prediction 0 and 1 when the smallest standardised coefficient exceeds a threshold, testing both constant thresholds and thresholds that depend on the number of bits left to predict. Empirically, this method was sub‑optimal relative to our proposed rule.
>
> *it would help me to understand how sensitive the method is to different sparsity levels. Did you observe cases where it becomes less reliable?*
>
> We tested stepwise and dual stepwise regression extensively across different Hamming weights in all four settings. For $n=256$, $\log q=20$ and 20M data, linear regression breaks at 42 cool bits, stepwise at 55 cool bits and dual stepwise at 70 cool bits. Above these thresholds, the method recovers 0/20 secrets respectively.
> We additionally manually inspected when the dual stepwise method fails as $h$ increases. For $n=256$, $\log q=20$ and 42 cool bits (where the linear regression already fails), the dual stepwise method is able to recover the cool bits in 20 out of 20 secrets. For $n=256$, $\log q=20$ and 52 cruel bits, we checked at which step the method misclassifies a bit (0 vs 1). In the six cases when our method fails, it has **16 or fewer bits left to distinguish**. The instability appears only at the very end.
>
> *Did you try any other regression variants (like ridge or LASSO) before settling on the stepwise approach, or is that left for future work?*
>
> We ran preliminarily L1 and L2 penalties, which gave results similar to plain linear regression. For L2, we believe this is because we still invert $A^TA + \lambda I$, effectively modelling feature correlations that, in this problem, do not exist. Moreover, we report that tuning the regularization coefficient for L1 and L2 was particularly challenging.
>
> **Broader perspective**
>
> *The paper does not discuss when the demonstrated recovery capabilities (e.g., number of cruel bits or Hamming weights achieved) would translate into practical cryptographic impact. A brief impact statement or discussion of real-world relevance would help readers interpret the results. [...] At what concrete success threshold (e.g., number of recovered cruel bits, Hamming weight, or % of secret recovered) would you consider this attack a practical threat to an LWE-based scheme in deployment?*
>
> In general, cryptographic attacks on public key cryptography advance (over the decades) by introducing new methods, scaling them up, introducing hard challenges (like RSA-512, RSA-768, RSA-1024, …) and keeping track of the resources used to solve these challenges.
>
> For PQC cryptography, we have not had the luxury of decades of research on attacks on the standardized parameters: for lattice-based cryptography where our paper is focused, LWE schemes were standardized by NIST in the period of 2017-2022, but using more risky parameter choices which had not been studied in the lattice reduction community for decades, namely the standardized schemes (Kyber and Homomorphic encryption) use small secrets (binary, ternary, and binomial) and very narrow error distributions. In addition, the Homomorphic Encryption Standardization (HES) community has pushed for allowing sparse secrets (very few non-zero entries) and the recent proposals for HE schemes propose using extremely sparse secrets [https://eprint.iacr.org/2025/784.pdf, https://eprint.iacr.org/2025/651.pdf].
>
> The AI-based attack studied in this work are based on the new-ish direction introduced in the SALSA and Cool and Cruel papers (2021-2024), and we are now in the stage of scaling them up and setting hard challenges such as those introduced in the LWE Benchmarking challenges. Our results do not currently break standardized schemes, but continue to make progress to scale up these solutions and hypothesize a scaling law which helps to understand what to expect from these methods *without any further ML innovations*.
>
> *how do your results translate to parameter choices in deployed LWE schemes? Do they suggest specific ranges that might be unsafe under these attacks?*
>
> Based on the experience of the crypto community over decades, and progress on these AI attacks, we do not recommend use of sparse secrets. Furthermore, moving to random secrets for LWE would mitigate these attacks and provide a wider margin of safety.

---

### Meta-Review · Area_Chair_zp8J · 2026-01-07

**Summary:**

the paper received mitigated responses from informed reviewers (the only reviewer championing the paper with a 8 acknowledges lack of familiarity and the use of an LLM to assess the paper),
reviewer eY9Z's concerns on suitability for an ML audience remain unaddressed (cf their follow-up on the rebuttal) despite clarifications,
reviewer ZUoz comment on cruel bits assumptions reveal a lack of clarify in the paper's introduction of the entire stack, the authors should take into account the remark by the reviewer (and their own response) when rewriting the paper either for the camera ready (if accepted) or for a next submission.

**Reviewer Concerns:**

several concerns raised by the reviewers were properly addressed by the authors,
reviewer eY9Z's concerns on suitability for an ML audience remain unaddressed (cf their follow-up on the rebuttal) despite clarifications,
reviewer ZUoz comment on cruel bits assumptions reveal a lack of clarify in the paper's introduction of the entire stack, the authors should take into account the remark by the reviewer (and their own response) when rewriting the paper either for the camera ready (if accepted) or for a next submission.

**Reviewer Scores:**

reviewer xzek acknowledges the educated guess (and the use of an LLM) in their review, they would not have changed their mind after discussion

reviewers eY9Z, 1oft and ZUoz

---

### Decision · Program_Chairs · 2026-01-26

Reject